# Extracellular Vesicles in Therapeutics: A Comprehensive Review on Applications, Challenges, and Clinical Progress

**DOI:** 10.3390/pharmaceutics16030311

**Published:** 2024-02-22

**Authors:** Jiyoung Goo, Yeji Lee, Jeongmin Lee, In-San Kim, Cherlhyun Jeong

**Affiliations:** 1KHU-KIST Department of Converging Science and Technology, Kyunghee University, Seoul 02447, Republic of Korea; t16139@kist.re.kr; 2Chemical and Biological Integrative Research Center, Korea Institute of Science and Technology, Seoul 02792, Republic of Korea; 219302@kist.re.kr (Y.L.); 220583@kist.re.kr (J.L.); 3KU-KIST Graduate School of Converging Science and Technology, Korea University, Seoul 02841, Republic of Korea; 4Division of Life Sciences, Korea University, Seoul 02841, Republic of Korea; 5Division of Bio-Medical Science & Technology, University of Science and Technology (UST), Seoul 02792, Republic of Korea

**Keywords:** extracellular vesicles, EVs, sEVs, exosome, EV application, EV engineering, large scale production, EV purification, clinical trial

## Abstract

Small Extracellular Vesicles (sEVs) are typically 30–150 nm in diameter, produced inside cells, and released into the extracellular space. These vesicles carry RNA, DNA, proteins, and lipids that reflect the characteristics of their parent cells, enabling communication between cells and the alteration of functions or differentiation of target cells. Owing to these properties, sEVs have recently gained attention as potential carriers for functional molecules and drug delivery tools. However, their use as a therapeutic platform faces limitations, such as challenges in mass production, purity issues, and the absence of established protocols and characterization methods. To overcome these, researchers are exploring the characterization and engineering of sEVs for various applications. This review discusses the origins of sEVs and their engineering for therapeutic effects, proposing areas needing intensive study. It covers the use of cell-derived sEVs in their natural state and in engineered forms for specific purposes. Additionally, the review details the sources of sEVs and their subsequent purification methods. It also outlines the potential of therapeutic sEVs and the requirements for successful clinical trials, including methods for large-scale production and purification. Finally, we discuss the progress of ongoing clinical trials and the implications for future healthcare, offering a comprehensive overview of the latest research in sEV applications.

## 1. Introduction

With the development of drug delivery, many nanoparticles (NPs) have been studied as carriers. Extracellular Vesicles (EVs) have gained prominence in drug delivery due to their intrinsic production, although they often require purification and potential modification for specific applications. EVs are categorized into exosomes, microvesicles, and apoptotic bodies based on biogenesis, ranging in size and composition [1]. Especially, sEVs, currently defined as having a diameter of approximately 30–150 nm, are small vesicles produced inside cells called multivesicular bodies (MVBs) [2,3]. They are formed when MVBs undergo a process called inward budding. This process results in the formation of intraluminal vesicles (ILVs) inside MVBs, which contain cytosolic components, including RNA, proteins, and lipids, from the cell [4,5]. Some of these ILVs are released into the space outside the cells through fusion with the cell membrane and are referred to as sEVs [6]. Others are transported to lysosomes for degradation [7,8].

During the inward budding process, certain proteins are incorporated into the membrane invagination within MVBs. This transfer of molecules allows sEVs to facilitate communication between cells without direct contact [9]. They can also alter the function or differentiation of target cells. They are released into the space outside the cells by fusion with the cell membrane. sEVs contain various molecules, including RNA, lipids, and proteins, that are transferred to other cells [10].

The characteristics of sEVs arise from their origins, which implies that sEVs could be representative of ‘an avatar of cell’ [11]. Additionally, researchers have observed that sEVs can accommodate functional molecules, such as proteins, DNA, RNA, and chemicals; this aspect of sEVs has emerged as a potential carrier [12]. Because sEVs inherit their parent cells, no external factors are required to load bio-substances such as intrinsically folded proteins and nucleic acids.

sEVs contain specific membrane proteins that identify the physiological and pathological states of the cells from which they originate or indicate their preferred target cells [13]. These membrane proteins can also be used or modified to allow sEVs to function as drug delivery systems and therapeutic platforms, including targeted therapeutic approaches. Compared to other nanoscale particles, sEVs have a strong advantage in membrane engineering because they consist of lipid bilayers that are capable of modification, similar to their parental cells. Consequently, sEVs have garnered much attention in recent years as a tool, not only as a cargo but also as a modifiable platform for membrane proteins, and several clinical trials have been established.

Despite the vast potential of sEVs for therapeutic applications, a number of obstacles must be overcome, such as difficulties in large-scale production, concerns about purity, and the lack of established protocols and characterization methods. The primary challenge in using sEVs for medical purposes is the requirement for mass production while maintaining their purity for industrial use. Therefore, most clinical trials to date have been conducted using cell-derived, unmodified EVs. EV heterogeneity presents another hurdle in this field; therefore, a thorough understanding of the characteristics of each EV is required.

This review discusses the potential of sEVs as vehicles for transporting functional molecules, including proteins, genetic materials, and chemicals. We also outline the requirements for successful clinical trials, including large-scale production using diverse sources and the adoption of advanced purification methodologies. Furthermore, we will highlight emerging clinical trials and suggest ways to develop engineered sEVs for clinical applications.

## 2. Therapeutic Strategies with EVs

EVs have been developed to treat diseases using the unique properties of intact EVs derived from various cell types. Research has been actively conducted on methods that can be engineered as a platform for targeted therapeutics and drug delivery. Aims to bridge the gap between the fundamental understanding of EV biology and its emerging role in clinical interventions, highlighting the recent advancements and ongoing challenges in this rapidly evolving field. In this section, we will first look at the therapeutic applications of intact EVs and then at EVs as therapeutic agents themselves or as drug delivery carriers through various forms of engineering.

### 2.1. Intact EV Therapeutics (X-Derived EV)

EVs, which contain various molecules derived from parental cells, have the potential to alter the characteristics of recipient cells and exert therapeutic effects (Figure 1). Therefore, EV injection may be a safer and more competitive option for therapeutic use than direct cell injection [14,15,16].

#### 2.1.1. Originated from Stem Cells

Stem cells have been used in cell therapies for regenerative diseases owing to their pluripotency and self-renewal properties to replace injured tissue [17]. Numerous studies have demonstrated that the principal therapeutic efficacy arises from EVs that contain bioactive molecules. As a result, stem cell-derived EVs have been explored for their potential in cell-free therapy due to their demonstrated capabilities [18].

Embryonic stem cells (ESCs) derived from the inner cell mass of blastocyst stage embryos maintain extensive proliferative activity under certain conditions [19]. ESCs produce large quantities of EVs [20]. As the ESC microenvironment has been reported to reprogram tumor cells less aggressively, ESC-derived EVs (ESC-EVs) can reduce cancer cell growth and tumorigenicity owing to the transfer of their cargo, including SOX2, OCT4, and NANOG proteins [21,22]. In addition, ESC-EVs induced glioblastoma cell apoptosis in an in vitro GBM model and in vivo subcutaneous and xenograft models [23]. In myocardial infarction, ESC-EVs enhance cardiomyocyte survival and neovascularization and modulate CPC-based repair programs in the heart [24]. However, human ESCs (hESCs) cause the destruction of human embryos, making hESCs research ethically difficult [25].

Mesenchymal stem cells (MSCs) isolated from adipose tissue, bone marrow, and other sources are spindle-shaped plastic-adherent cells [26]. MSCs are thought to be the most suitable candidates for tissue regeneration because of their ability to differentiate, proliferate, and modulate inflammation, suggesting that MSC-derived EVs (MSC-EVs) would exhibit similar functions [27,28]. For example, bone marrow MSC-EVs have low immunogenicity [29]. MSC-EVs are aimed at treating ARDS, liver failure, sepsis, and cutaneous wound healing [30,31]. However, as the passage number of MSCs in cell culture increases, cell activity and EV production decrease [32,33].

Induced pluripotent stem cells (iPSCs) are induced by somatic cells, which are reprogrammed directly by ectopic expression of transcription factors (OCT4, SOX2, KLF4, and MYC) related to embryogenesis [34]. iPSCs have comparable abilities to ESCs but are free of ethical issues. iPSC-derived EVs (iPSC-EVs) promote EC tube formation and microvessel sprouting to recover myocardial cells without increasing the frequency of arrhythmogenic complications [35]. Moreover, iPSC-EVs improved cell migration, proliferation, and tube formation to promote angiogenesis in HUVECs [36]. iPSC-EVs are effective in wound healing treatments as they promote fibroblast migration [37]. However, the risk of tumor formation remains a problem that must be resolved using iPSCs [38].

Stem cell-derived EVs have been used for tissue regeneration, wound healing, and anticancer. However, each stem cell type has a clear weakness, and researchers are attempting to overcome this obstacle. A prime example is iPSCs derived from MSCs, referred to as iPSC-MSCs. iPSC-MSCs are less tumorigenic and exhibit self-renewal potency, despite several passages [39,40,41]. iPSC-MSC-derived EVs are beneficial for wound healing, hepatoprotection, and ischemic disease treatment [33,42,43].

#### 2.1.2. Originated from Immune Cells

The immune system comprises various immune cells, each specializing in a unique role. Granulocytes such as neutrophils, eosinophils, and basophils play a key role in neutralizing pathogens by releasing antibacterial substances. Antigen presenting cells (APCs), like dendritic cells and macrophages, take up the task of processing and presenting antigens. Lymphocytes, including T, B, and NK cells, are integral to adaptive immunity [44]. In a parallel perspective, EVs derived from immune cells carry a distinct composition for immunomodulation [44]. This has generated substantial interest and development in the field of EVs associated with immunocompetent cells, making it a significant domain in EV application research [45].

Dendritic cells (DCs), often referred to as ‘unprofessional’ antigen-presenting cells (APCs), excel in their ability to acquire and present antigen molecules to T cells, which is a vital aspect of immune response regulation [46,47]. Therefore, they play critical roles in the development of therapeutic strategies for immune-mediated conditions. A particularly intriguing aspect of DCs is their derived EVs. These EVs not only incorporate conventional EV markers but are also enriched with key proteins involved in antigen presentation, such as major histocompatibility complex (MHC) class I or II molecules and the co-stimulatory molecule CD86, which are crucial components of immune response modulation [45,46].

DC-derived EVs (DC-EVs) containing proteins related to antigen presentation are used in antigen-specific immunotherapy. Specifically, matured DC-EVs, rich in HLA-DR, MHC Class I, CD40, and CD80 compared to their immature counterparts, stimulate CD8 T cells to induce IFN-γ production [47]. DC-EVs with a high concentration of MHC class I molecules can also activate CD8 T-cell hybridomas [48]. Researchers are currently interested in using DC-EVs as cell-free vaccines. EVs derived from tumor peptide-pulsed DC can prime specific cytotoxic T lymphocytes; therefore, DC-EVs can reduce murine tumor growth in a T cell-dependent manner [49].

Initially, researchers anticipated that DC-EVs could directly present antigens to T cells. However, recent findings suggest that DC-EVs can indirectly activate T-cells [50]. For example, DC-EVs containing peptide-MHC II complexes can be delivered to MHC class II-deficient DCs, and these recipient DCs can then promote antigen-specific CD4 T cell activation [51,52]. One of the primary reasons why DC-EVs fail to directly activate T cells is their low antigen-presenting capacity. To address this issue, second-generation DC-EVs, which are derived from IFN-γ-matured DCs and express MHC class II and CD86 molecules abundantly, can enhance T cell priming [53]. In a phase II clinical trial for inoperable non-small-cell lung cancer, second-generation DC-EVs showed impressive immunotherapeutic effects by boosting NK cell function in patients [54].

The other ‘Professional’ APCs are the B cells that include antigen and produce antibodies and present antigen to CD4 T cells by MHC class II [55,56]. Although B cell-derived EVs are known to activate CD4 T cells by antigen-presenting MHC class II, they induce CD4 T cell apoptosis due to FasL [57,58,59]. Especially, EVs derived from B cells transformed by Epstein–Barr Virus (EBV), which are commonly used for large-scale EV production, have been found to have a high prevalence of FasL [5,60]. Additionally, these EVs contain the viral latent membrane protein (LMP1) and EBV-miRNA, which suppress EBV target genes by being conveyed to DC [59,60].

In conclusion, there is an urgent need to enhance the therapeutic efficacy of both DC- and B-cell-derived EVs. The production of DC- and B-cell-derived EVs involves considerable challenges, including the intricate process of establishing cell lines. Although strides have been made with the advent of second-generation DC-EVs to address the limitations of antigen presentation [61,62], it is imperative to develop assays that can quantify MHC class II proteins at the single-molecule level. The presence of viral components in these EVs, as observed in EBV, raises concerns regarding their potential role in disease progression and the spread of viral information. This necessitates further research on the dual nature of these vesicles as potential therapeutic agents and carriers of pathogenic signals. Such advancements will not only ensure better exploitation of the therapeutic potential of EVs but also drive the frontier of personalized immunotherapies.

#### 2.1.3. Originated from Tumor

TEVs, EVs derived from tumors, suppress immunity and allow the advancement of cancer in the tumor microenvironment [63,64]. These TEVs have been observed to transform MCF10A cells into tumor cells, a process associated with an increase in mature miRNAs, which correlates with the concentration of dicer in EVs [65]. Particularly, TEVs containing ΔNp73 mRNA promote tumor growth and increase drug resistance [66]. In colon cancer patients, a high level of ΔNp73 mRNA in EVs is linked with shorter disease-free survival, found in approximately 30% of cases.

Conversely, TEVs laden with various tumor antigens are emerging as viable candidates for cancer vaccines. These TEVs inhibited cancer growth and promoted CTL-mediated antitumor immunity [67,68]. They transfer accumulated tumor antigens to DC, which induces a potent CD8 T cell-dependent anticancer effect [65].

Moreover, heat-shocked tumor-derived EVs load a large amount of tumor antigen peptides such as HSP 70 and Hsp 90, which can trigger immune responses so that they can resist tumor cells by immune cells [69,70]. Another strategy for loading HSP 70 into TEVs to stimulate DC maturation is to express membrane protein-bound HSP 70 [71].

It has been suggested that the direction of immune activation or suppression may depend on the type of stimulation when the immune system interacts with TEVs [72]. For example, the concentration of TEVs interacting with T cells determines the fate of the T cells [73]. Glioma-derived EVs increase T cell proliferation, lymphocyte development, lymphocyte differentiation, and migration functions at low concentrations (100 μg/mL), but increase lympho-hematopoietic cancer, lymphomagenesis, and hematological neoplasia functions at high concentrations (2000 μg/mL). Therefore, caution should be exercised when using TEVs for vaccination.

### 2.2. Engineered EVs for Protein Delivery

EVs are used as carriers to deliver proteins because they contain proteins from the parental cells. Therefore, the most representative method is to overexpress a specific protein in the parental cells so that it can be loaded into EVs in a concentration-dependent manner. However, proteins that are difficult to express inside the cytoplasm, such as transcription factors, are difficult to carry into EVs by overexpression alone; thus, various platforms are being developed.

In addition, unlike other nanoparticles, EVs are the only carriers that can deliver intact membrane proteins to recipient cells. Therefore, studies aimed at maximizing the therapeutic efficacy of engineering EV membranes are being actively conducted (Figure 2).

#### 2.2.1. Membrane Protein Delivery

Membrane proteins play essential roles in many physiological processes involved in signaling and cell recognition [74]. Liposomes and EVs are commonly used to deliver proteins to membranes. Liposomes must undergo a series of complex manufacturing processes to express membrane proteins and cannot provide intact membranes [75]. Unlike liposomes, EVs can be used to express membrane proteins by transfection and can provide an intact membrane environment because they are made from cells. Therefore, membrane proteins can deliver a more intact structure with EVs than with liposomes.

EV marker proteins, such as LAMP2b, ALIX, and TSG101, associated with multivesicular bodies (MVB), as well as CD63, CD81, and CD9 from the tetraspanin family, are abundant in EVs and can be used to enrich specific proteins of interest [1,76]. These markers are instrumental to enriching specific proteins of interest. Efficient sorting and accumulation in EVs can be achieved by conjugating these marker proteins to the protein of interest. LAMP2b is a member of the lysosome-associated membrane protein (LAMP) family located in endosomes and lysosomes. The specific peptides or proteins conjugated to the N- terminal of LAMP2b are able to be displayed on the EV surface [77,78,79]. However, since LAMP2b undergoes rapid degradation through deglycosylation before glycosylation, glycosylated LAMP2b is recommended for the stable expression of N-terminal conjugated proteins [80,81,82].

Members of the tetraspanin family, which feature both N- and C-terminals in the cytosol, can encapsulate soluble proteins within EVs. Moreover, modifying the deletion of domains associated with a large extracellular loop for the insertion of desired proteins is possible [83,84,85,86,87]. PTGFRN and BASP1 have recently been identified as EV marker proteins that facilitate transfer to EVs [88].

The pDisplay vector has been widely used as a vector capable of expressing desired proteins on the surface of EVs using the transmembrane domain of the platelet-derived growth factor receptor (PDGFR) [89,90,91]. Although not described as an EV marker protein, it is known to be well transferred into EVs by transfection.

Previous research has established that EVs tend to accumulate in organs such as the liver, spleen, and kidneys [92,93,94,95]. This natural propensity makes them particularly effective in treating liver-related conditions such as acute liver failure, even in the absence of specific targeting peptides [96,97,98]. However, a targeted approach is essential for directing EVs to specific organs.

To achieve targeted delivery Peptides, such as rabies virus glycoprotein (RVG), GE11, and nanobodies, have been used [77,89,99,100]. In particular, they can transport ligands to the brain by interacting with acetylcholine receptors overexpressed in neuronal cells [77,99,101,102]. When conjugated to LAMP2b, RVG peptides effectively deliver microRNAs (miR-124) and small interfering RNAs (siRNA) within EVs to the brain [77,99]. This delivery system has the potential to treat neurological conditions and improve the efficacy of EV therapies.

Building on the concept of targeted EV therapy, GE11 peptides were identified as specific binders to epidermal growth factor receptors (EGFR) through phage display peptide library screening [100]. This specificity is particularly significant in breast cancer cells with high EGFR expression levels. When GE11 peptides are present in EVs loaded with miRNAs, they can selectively target and treat breast cancer cells [89]. Furthermore, the use of EGFR nanobodies displayed on the surface of EVs enhanced their binding affinity for breast cancer [103].

Expression of biologically active proteins on the surface of EVs is a powerful tool for inducing direct therapeutic effects. These proteins can be expressed in various ways, not only in a concentration-dependent manner but also through diverse platform-based approaches, expanding the potential applications of EV-based therapies [86,90,91,104,105,106,107,108,109,110,111].

One notable example is IFN-induced transmembrane3 (IFITM3), known for its antiviral properties [112]. EVs expressing IFITM3 have demonstrated the ability to suppress Zika infection in pregnant mice and their fetuses [109], as well as dengue virus infections [110]. The virus-neutralizing capabilities of EVs, particularly those expressing proteins that interfere with viral cell entry, have garnered significant interest. This is particularly evident in studies focused on combating the global spread of SARS-CoV-2, where EVs containing ACE2, which interact with the viral spike protein, act as decoy receptors. These ACE2-bearing EVs prevent the virus from binding to ACE2-positive cells, thus offering a novel approach for protection against SARS-CoV-2 infection [111,112].

In cancer therapy, the delivery of membrane proteins via EVs is challenging. Tumor cells often evade the innate immune system by expressing CD47, a ‘do not eat me’ signal that inhibits phagocytosis when it binds to SIRPα on macrophages, thereby promoting tumor invasion [113]. EVs containing CD47 evade macrophage phagocytosis and induce ferroptosis by delivering drugs to tumor cells [86]. Conversely, SIRPα-containing EVs can enhance phagocytosis by blocking CD47 in tumor cells [90]. The glycoprotein of the vesicular stomatitis virus (VSVG) is the viral membrane protein that can serve as an attachment and have a fusion effect at low pH [114]. Xenogenized tumor cells fused with VSVG-containing EVs allow DCs to recognize the tumor [109]. VSVG-containing EVs can be efficiently delivered Cas9-sgRNA ribonucleoproteins (RNPs) to edit target cells genomes [115,116,117,118].

Furthermore, the expression of antibodies targeting two different cells on the surface of EVs can increase the likelihood of cell–cell interactions [111]. Using this approach, they developed synthetic multivalent antibody-retargeted exosomes (SMART-Exos). For instance, SMART-Exos containing monoclonal antibodies CD3 and EGFR can induce cross-linking between T cells and EGFR-positive breast cancer cells [91]. Another variant of SMART-Exos, equipped with anti-human CD3 and anti-human HER2 antibodies, can connect HER2 positive breast cancer cells and T cells, providing a chance to kill tumor cells directly [111]. This innovative use of EVs in targeted therapy opens up new avenues for the treatment of various diseases, particularly in oncology.

#### 2.2.2. Soluble Protein Delivery

EVs composed of a lipid bilayer can stably maintain internal materials even when exposed to serum, making them ideal for protein delivery [119]. This stability increases the half-life of proteins encapsulated within the EVs. There are two primary strategies for delivering proteins of interest to EVs: concentration-dependent delivery and active sorting delivery approach [83,84,120,121,122,123,124,125].

Concentration-dependent delivery involves protein overexpression in the cytoplasm through transfection, which induces excessive expression to allow partial loading during EV biogenesis based on the concentration. For example, macrophages transfected with plasmids encoding murine IL-10 secreted EVs containing IL-10, thus promoting M2-type polarization to prevent ischemic acute kidney injury [121]. This method is straightforward, but less effective for proteins located in the nucleus, such as transcription factors. They are primarily used as cytoplasmic proteins.

The active sorting delivery approach, which involves encapsulating the desired protein and tethering molecules to EVs, is more effective than passive loading. In this approach, many researchers have used proteins involved in EV biogenesis as the tethering molecules. The ESCRT complex, which is involved in EV biogenesis, contains subunits with ubiquitin-interacting motifs that facilitate the delivery of ubiquitinated proteins of interest into EVs [122]. Similarly, several antigens tagged at the C-terminus with ubiquitin can be encapsulated in EVs, which can potentially serve as novel vaccines [123]. The WW tag method is another approach used to ubiquitinate desired proteins in cells [124]. When the desired protein is conjugated to a WW tag, it interacts with the L-domain (late domain)-containing protein Ndfip1 (Nedd4 family interacting protein 1), which is present in endosomes as a ubiquitin ligase adaptor protein. The desired protein with a WW-tag can be recognized by Ndfip1, resulting in its ubiquitination and encapsulation in EVs. Gag proteins serving as the main determinant Retroviral particles assembly can conjugate Cas9-RNPs to incorporate in EVs [117].

The development of active sorting delivery approach within EVs has been advanced by using fluorescence-mediated dimerization, Exosomes for protein loading via the optically reversible protein–protein interactions (EXPLOR). EXPLOR method use tetraspanin proteins [84]. In EXPLOR, CD9, a tetraspanin protein, conjugates with CIBN (a truncated version of the CRY-interacting basic-helix-loop-helix 1), and the protein of interest conjugates with CRY2 (cryptochrome 2). The interaction between CIBN and CRY2, which is regulated by blue light-dependent phosphorylation, allows for transient control. Using this technique, the super repressor IκB has been loaded into EVs to reduce mortality and inflammatory response, and to lower the risk of premature birth in a mouse model of sepsis [83,125]. In addition, ligand-mediated dimerization by Rapamycin-induced FKBP-FRB can efficiently sort Cas9-RNPs into EVs [126]. These various dimerization techniques have been explored to compare the loading efficiency of Cas9-RNPs and verify effective gene editing activity [127].

Another approach is to mimic the manner in which viruses use EVs to evade the immune system [128,129,130,131]. For example, herpes simplex virus type 1 has been shown to results in a 100-fold reduction in stable cell lines expressing dominant-negative VPS4, which is essential for ESCRT function [128]. The LMP1 of the EBV can be controlled to load into EVs by CD63, a well-known EV biomarker protein [129]. Additionally, EVs secreted by cells infected with *M. tuberculosis* contain many mycobacterial proteins that are delivered into EVs through ubiquitination [130].

Particularly noteworthy is the potential to deliver the desired proteins into EVs if a specific domain of a viral protein interacts with an EV-related protein. The viral protein pX of hepatitis virus binds to domain 5 of ALIX [131]. These observations suggested that pX-GFP accumulates in EVs, indicating its potential as a useful platform for soluble protein delivery.

### 2.3. Engineered EVs for Non-Protein Delivery

Over the years, EVs have been widely used as mediators for delivering non-protein agents, such as drugs and nucleic acids [77,85,89,99,132,133,134,135,136,137,138,139,140,141,142,143,144,145,146]. A method for loading drugs and nucleic acids has been developed to reduce degradation, maximize low-dose activity, and minimize toxicity (Figure 3) [141]. In particular, the technology that shows therapeutic effects by delivering functional cargos through EVs has many advantages [77]. EVs have low immunogenicity and the unique ability to pass through biological barriers. Moreover, EVs can load biological molecules, both extrinsic and intrinsic, unlike liposomes, which have a structure similar to that of EVs [147]. In addition, targeting abilities can be increased through EV surface engineering to reduce nonspecific side effects [77,89,99,100,103].

#### 2.3.1. Chemical Drugs

EVs loaded with chemical drugs have been investigated for their therapeutic applications, unveiling substantial potential in the realm of medical treatment [132,133,134,146,148,149,150,151]. Incorporating chemical drugs into nanoparticles (NPs), such as EVs, can reduce side effects and maximize delivery efficiency. Engineering EVs for loading anticancer drugs is particularly important in cancer research [132,133,134,148].

Doxorubicin (Dox) and paclitaxel are common anticancer drugs with non-specific toxicities. Long-term exposure to Dox can lead to severe cardiotoxicity. Research on packaging these drugs using NPs such as EVs is being conducted to reduce side effects on non-target cells [136,152]. Dox, which can be loaded into EVs via simple incubation, is also easily released. Therefore, direct conjugation of Dox to EVs and loading them inside is being developed. For example, a pH-sensitive DNA carrier attached to an EV was used to transport Dox to the tumor environment, facilitating the release of intercalated Dox in response to acidic conditions present there [151]. In another study, Dox was released when the imine bond was disrupted under an acidic condition [153].

Through membrane engineering, small amounts of EVs can reduce nonspecific side effects and exert greater therapeutic effects [135,154,155]. One study reported that an iRGD peptide conjugated with LAMP2b was expressed on the EV membrane, enabling cancer-specific treatment with Dox [141]. To increase the targeting effect of Dox, an antibody against A33, overexpressed in colorectal cancer cells, was attached to the EV surface, leading to reduced off-target side effects [142]. Moreover, EVs can express functional enzymes such as PH20 on their membranes to decompose the tumor microenvironment [143].

#### 2.3.2. Nucleic Acids

Gene therapies using RNAs require delivery platforms, such as EVs and liposomes, because of RNA’s unstable structure and susceptibility to RNase degradation. EVs are particularly promising for RNA delivery because they can encapsulate RNAs and evade the immune system. RNA interference (RNAi), which induces decomposition and silencing after mRNA transcription, is a promising tool for gene therapy. RNAi serves to degrade mRNA, thereby inhibiting specific protein synthesis, whereas the direct delivery of mRNA can facilitate sustained protein production. Electrophoresis is commonly used to load exogenous siRNAs into EVs. For example, siRNA-loaded EVs have been used to deliver siRNAs across the mouse blood–brain barrier (BBB) [77]. To treat glioblastoma, ASO-21, which is complementary to miR-21, was electroporated into EVs [139]. In another study, an excessive amount of plasmid DNA was injected into hAdMSCs using a commercially available track-etched membrane (TM-nanoEP) to increase the amount of mRNA loading into EVs [156].

However, electroporation can alter EV morphology; therefore, transfection of small RNA plasmid vectors into parental cells is being developed as an alternative. Strategies to increase the EV loading efficiency include the use of expression plasmid vectors conjugated to LAMP2b [103]. For example, co-transfection of RVG-Lamp2b and Nerve Growth Factor (NGF) plasmids was performed to efficiently deliver NGF mRNA with neuroprotective functions to EVs for the treatment of ischemia [78]. Studies have also focused on loading RNAs using EV markers, such as CD9 and CD63 [89,157]. For example, miR-155 fused to CD9 was used to enhance the efficiency of miRNA loading into EVs [157]. Another study also showed that EVs modified using RVG-LAMP2b (targeting device), CD63-L7Ae (RNA packing device), and CX43 S368A (cytological delivery helper) proteins could efficiently deliver mRNA to treat Parkinson’s disease [158]. Several studies have shown that certain miRNA motifs are specifically loaded into EVs. In particular, the EXOmotif, which includes CGGGAG, is involved in exporting EVs, and a previous study reported that hnRNPA2B1 recognizes and selects the motif of miRNAs that are SUMOylated and directly loaded into the EV [159,160].

It is necessary to carefully select parental cells and express targeting proteins for the effective delivery of mRNA-loaded EVs. It offers advantages in terms of immune evasion and targeting when utilizing EVs extracted from cells of the same origin. In a study, it was observed that the delivery of lung-derived EVs (Lung-EVs) through nebulization resulted in longer retention in the bronchioles and parenchyma compared to HEK-EVs and LNP [161]. By engineering the surface of EVs with minimal side effects, many strategies to increase their targeting effect in nucleic acid delivery have been actively studied. To target cancer cells overexpressing EGFR, miRNAs were loaded into EVs released from parental cells with the overexpressed GE11 peptide to reduce side effects and increase treatment effects [89].

## 3. Large-Scale Production of EVs for Clinical Use

This chapter highlights the importance of selecting appropriate EV sources for clinical applications. It also underscores the challenges in EV production, setting the stage for deeper exploration of various EV sources and their unique attributes in the subsequent section.

### 3.1. Sources of EVs

EVs, as extracellular vesicles, display a range of biological properties that are significantly influenced by the type of parental cell. The selection of parental cells is a critical factor in defining the specific characteristics and functionalities of EVs and should be carefully aligned with their intended therapeutic or diagnostic applications. Despite the challenges in achieving robust yields, the demand for large quantities of EVs for clinical purposes underscores the need for effective scale-up strategies. This involves not only choosing the right source of EVs but also optimizing the culture methods to enhance yield while minimizing loss.

For clinical applications, it is imperative that EVs meet stringent criteria in terms of purity, biological properties, and detailed characterization. This level of quality assurance is essential to ensuring safety, efficacy, and consistency in therapeutic settings. In the following discussion, we delve into the characteristics of EVs derived from various sources and highlight how their origin influences their intrinsic properties and potential clinical utility. This exploration is crucial for understanding how to harness the unique qualities of EVs from different cell types and tailor their production for specific clinical needs.

#### 3.1.1. Human Cell Line

EVs vary in their unique characteristics depending on the biological characteristics of the parental cell. In addition, it is necessary to consider which cells are most efficient for use because the properties of EVs vary depending on the cell culture conditions.

MSCs are multipotent cells that proliferate, differentiate, and regulate immunity. MSC-exos are also known to have the ability to lower inflammation and regenerate tissue [157,158,162,163]. MSC-exos have low surface antigenicity, resulting in low immunogenicity and reduced immune rejection, making them safe, and have been extensively studied in clinical trials. However, scaling-up is challenging owing to the adherent nature of MSCs and concerns regarding the tumorigenic properties associated with TERT. Recent studies have focused on increasing the MSC-exo yield through 3D culture, improving the yield by up to 140 times [164]. Enhancements in yield have also been reported through the internalization of surface-modified positively charged nanoparticles, which accelerate autophagy [165]. Moreover, Myc transformation can be considered an alternative for immortalizing MSCs. Studies have reported that through Myc transformation, MSC properties are maintained while inducing rapid proliferation, leading to an increased production of EVs [166].

HEK293 cells are widely used in EV research owing to their high transfection efficacy and ease of modification. These cells can be engineered to express fusion-capable proteins on the EV surfaces to target specific cells [113]. Several drugs using HEK293 cells have received FDA approval for clinical trials [167]. In terms of production, HEK293 cells offer advantages for scale-up owing to their rapid cell division and easy cultivation. However, for adherent cells, optimizing the protocols to increase EV yield is crucial. Studies have shown significant increases in EV markers and protein concentrations under low-pH culture conditions [168].

#### 3.1.2. Milk EV

Milk EVs have recently emerged as a platform for the delivery of macromolecules. Milk EVs have been highlighted for their potential for mass production from abundant milk sources containing lipids, mRNAs, miRNAs, DNA, and proteins, similar to other mammalian EVs. Milk EVs express CD9, CD63, and CD81 on their surface and contain MGF-E8 and flotillin-1 as internal markers. They have been successfully isolated from various animals and maintain their phospholipid bilayer structure and biological activity even during refrigeration [169]. Many researchers have shown that cargoes, particularly RNAs, are not degraded, even during the industrial process of milk EV production [169,170]. Milk EVs have been proposed as effective oral delivery platforms because of their long circulating half-lives and lack of systemic toxicity. Studies have shown that Curcumin encapsulated in milk EVs has high intestinal permeability and stability [171]. However, expressing specific proteins in milk EVs without genetically modifying the primary production source of cows is challenging. Additionally, purifying milk EVs requires complex methods, such as high-speed centrifugation and size exclusion chromatography, complicating mass production [172]. However, these refining methods can complicate mass production.

#### 3.1.3. Plant EV

Plant-derived EVs are nanoparticles that offer an alternative to the technical limitations of mammalian vesicles, with ongoing research on their suitability for various applications. The extract yield of plant-derived EV from grapes is 1.76 mg/g, grapefruit is 2.21 mg/g and tomato is 0.44 mg/g, respectively [173]. These high-yield plant-derived EV production sites are promising candidates for clinical use. Moreover, many studies have reported that plant-derived EVs have a similar size distribution, charge, shape, and components to mammalian-derived EVs, and clinical trials are in progress (NCT04879810, NCT04698447, NCT01668849) [174]. In addition, it has low toxicity, good biocompatibility and stability, and is edible. Therefore, an appropriate plant-derived EV can act as a natural therapeutic agent against various diseases [175].

Recently, several studies on the potential of plant-derived EVs as therapeutic agents have been published. Ginger-derived nanoparticles protect against alcohol-induced liver damage by increasing the expression of liver detoxification and antioxidant genes through the activation of Nrf2 and are selectively absorbed by intestinal macrophages to improve colitis [164,176]. In addition, several studies have demonstrated the anti-inflammatory and anticancer effects of plant-derived EVs.

However, research on the specific substances within plant-derived EVs that confer therapeutic effects is still in its infancy. As a result, there is a potential risk of unexpected immune responses or reactions owing to unknown substances present in these EVs. This highlights the need for further research and cautious application in therapeutic contexts.

### 3.2. Purification

EVs produced by cells include not only EVs but also apoptotic bodies and microvesicles. However, in the serum, EVs are mixed with high levels of LDL, HDL, and VLDL, necessitating a purification process to obtain pure EVs. This section briefly explains commonly used purification methods.

#### 3.2.1. Ultracentrifugation

Ultracentrifugation is widely used for EV isolation. The basic principle is to separate EVs and impurities according to their density and viscosity by high-speed centrifugation at a high speed (100,000× *g* or more) [165]. However, the amount of EVs that can be centrifuged simultaneously is limited when ultracentrifugation is used to purify EVs on a large scale. If the EVs are centrifuged for more than 4 h for purification, they may become contaminated with soluble proteins [167]. Ultracentrifugation can also cause sample aggregation, leading to a decrease in EV yield.

To avoid these issues, researchers often redissolve the EV pellet in PBS and then perform density gradient ultracentrifugation to further purify the EVs. Density gradient ultracentrifugation (DGU) uses sucrose or iodixanol to separate materials based on density. However, this method limits the sample volume that can be loaded onto a gradient. In addition, reproducibility may vary depending on the operator, and the use of high sucrose concentrations may cause sample contamination.

#### 3.2.2. Size-Exclusion Chromatography

Size-exclusion chromatography separates EVs from other EVs based on their size. Small molecules penetrate the pores of porous particles, whereas larger molecules take a shorter path and elute faster. This method results in high-purity isolated EVs with a lower risk of aggregation and shear stress than ultracentrifugation [177,178]. However, it is challenging to process large-scale samples, and the concentration of EVs may decrease after chromatography.

#### 3.2.3. Tangential Flow Filtration (TFF)

Tangential Flow Filtration (TFF) is a filtration method that addresses some of the limitations of traditional dead-end filtration, such as the ‘filter cake’ phenomenon, which can reduce filtration efficiency. This issue arises in dead-end filtration because the sample and filtration flows are in the same direction, leading to the accumulation of filtered particles on the filter surface. In contrast, TFF operates in a sample flow direction perpendicular to the membrane surface. This orientation allows for continuous diafiltration and results in efficient purification of high-purity EVs.

One of the key advantages of TFF is its ability to purify large-scale EV samples in a time-effective manner [179]. Unlike ultracentrifugation, TFF does not subject EVs to high-speed forces that can potentially damage them or reduce their yield. Therefore, TFF is a mild method for the isolation of EVs, helping to maintain their integrity and improve the overall yield [180,181].

Given that EVs share size and density characteristics with other extracellular vesicles, relying solely on density- or size-dependent purification methods may not yield completely pure EVs. Consequently, the combination of both techniques is often considered the gold standard for EV purification. This approach is particularly crucial for EVs intended for clinical trials, where Good Manufacturing Practice (GMP)-grade purification is essential [58,182,183].

For optimal purification, we recommend employing size-based TFF alongside DGU. TFF is suitable for large-scale purification and is capable of removing both large vesicles and small molecules based on size while concurrently concentrating the sample. At the same time, the DGU can help eliminate additional contaminants by addressing the concentration limitations posed by the dead volume in the TFF column. Collectively, these processes enhanced the yield and purity of EVs more effectively than differential centrifugation alone.

## 4. Clinical Trials Involving EVs

Unlike other synthetic nanoparticles, EVs have been studied as biomarkers for various diseases because they are intrinsically secreted nanoparticles. In clinical trials involving EVs, a substantial proportion—nearly 80%—focuses on their use for diagnostic and prognostic purposes.

Therapeutic research on EVs is gaining momentum. Currently, most of the clinical trials for therapeutic EVs involve naturally occurring, unmodified EVs, which are designated as ‘X-derived EVs’ in Table 1. Most clinical trials have been conducted on diseases aimed at immunosuppression. In particular, MSC-exos are known to contain immunosuppressive molecules such as anti-inflammatory cytokines and miRNAs. They are being utilized in treating conditions like SARS-CoV-2 and ARDS, leveraging their immune-suppressive capabilities, and are also applied in various infectious diseases. [NCT04491240], [NCT04602442], [NCT04798716], [NCT05216562], [NCT04657458], [NCT04493242], [NCT05116761], [NCT05125562], [NCT04276987], [NCT04602104]

Clinical trials involving immune cell-derived EVs are extremely rare, with only a few studies focusing on EVs derived from dendritic cells (DCs) and T cells. [NCT01159288], [NCT04389385] For DC-derived EVs, DC-EVs, a challenge has been the limited loading of TAA-MHC II complexes, which are essential for activating other immune cells. Recently, various strategies for accumulating TAA-MHC II complexes in EVs have been developed, and clinical trials are in progress using EVs secreted by IFN-γ-stimulated DCs as second-generation DC-EVs. [NCT01159288] It is known that the yield of T cell-derived EVs is significantly low. As a large number of EVs must be used in clinical trials, it is important to exploit technologies for the mass production of T cell-derived EVs.

However, few clinical studies have investigated plant-derived EVs. [NCT01668849], [NCT04698447], [NCT04879810], [NCT03493984] This is primarily attributed to the absence of an established, optimal protocol for their purification, compounded by the limited availability of plant-derived mediums. Information regarding the size, composition, and therapeutic efficacy of biomaterials in plant-derived EVs is limited. To reduce these side effects, plant-derived EVs should be characterized.

EVs have attracted attention owing to their therapeutic applications, particularly in protein delivery. Proteins expressed and loaded inside cells maintain their original structure and activity when carried by EVs. Proteins expressed on the cell membrane can also be expressed on the EV membrane, enabling targeted treatment and efficient delivery. The expression of fusogenic proteins on EVs enhances delivery efficiency by facilitating the delivery of internal substances into target cells.

Currently, only four clinical trials have investigated the use of engineered EVs. These include studies targeting SARS-CoV-2 or Sepsis with EVs overexpressing the CD24 protein [NCT04969172], [NCT04747574], a study on pancreatic cancer treatment using KRAS siRNA-loaded EVs [NCT03608631], and another on colorectal cancer treatment using curcumin-loaded plant-derived EVs [NCT01294072]. This limited number of trials reflects the existing challenges in transitioning engineered EVs from research to therapeutic applications.

Although EVs are still in the early stages of clinical trials, they hold great promise as versatile and innovative therapeutic platforms. Considered first-generation EVs, X-derived EVs that use only cellular properties should be moved toward engineered EVs as the second generation.

To facilitate this development, it is crucial to establish characterization assays, particularly at the single-molecule level, to confirm the presence and efficacy of these effector substances. Therefore, it is necessary to develop characterization assays for single molecules.

## 5. Discussion

EVs, naturally occurring nanoscale vesicles secreted by virtually all cell types, have emerged as a significant focus in the field of biomedical research because of their intrinsic role in intercellular communication and their vast potential in diagnostics and therapeutics. These small but complex structures encapsulate a variety of biological molecules, including proteins, lipids, and nucleic acids, that reflect the physiological state of their parent cells and can influence the behavior of recipient cells. The versatility of EVs has been harnessed for a multitude of applications, from serving as biomarkers of disease state and progression to acting as vehicles for targeted drug delivery, offering a promising avenue for the development of novel, minimally invasive therapies.

We explored the complicated nature of EVs as therapeutic entities, emphasizing their derivation from various cell types, including stem and immune cells. We highlight the potential of stem cell-derived EVs for regenerative medicine and immunomodulation, as well as the role of immune cell-derived EVs in cancer treatment. Given that the functional attributes of EVs are inherently linked to the properties of their source cells, the selection of appropriate parental cells is critical for harvesting EVs with the desired therapeutic functions. Cell modifications, such as transfection, are essential for the use of engineered EVs. Thus, establishing a stable cell line, rather than a transient one, to achieve a uniform amount and quality of EVs is crucial. Furthermore, one should be aware of possible problems, such as EV heterogeneity, uneven amounts produced, and virus-infected cell lines, in manufacturing EVs from a certain cell type. The development of a cell line that is easy to handle and has a high yield must be considered.

EVs, which are naturally secreted from cells, are the most endogenous type of NPs that have been developed as carriers for delivering functional cargo in an intact membrane environment compared to other NPs. Membrane proteins and loaded cargos can be produced in a more delicate and complex manner by living cells than by synthetic methods; however, we still need to establish a strategy for maximizing active substances because there are limitations in the cost and time required for EV production. These intrinsically maximized engineered EVs can effectively deliver functional cargo and remain intact in recipient cells. When cells endocytose EVs, the loaded substances can either be digested via lysosomal fusion or escape from the endosome. Only substances that escape endosomal compartments can function; therefore, researchers may need to devise methods for endosomal escape or cell membrane fusion.

The membrane proteins of EVs have mainly been studied for their targeting. These intact membrane proteins can also function in other ways, such as effective blocking, by forming clusters that other nanoparticles cannot. Therefore, researchers should consider the potential uses of these membrane proteins. Even when using membrane proteins for in vivo targeting, it is important to thoroughly discuss ways to target specific cells and organs, consider the most specific proteins for targeting, and prevent off-target effects. It is also necessary to confirm the cells that take up EVs and their biodistribution.

Next, we address the challenges and strategies for large-scale EV production for clinical use. Different sources of EVs, including human cell lines, milk, and plants, were investigated, and their potential for scalable manufacturing was examined. We underscore the importance of ensuring EV purity and functionality, reflecting on the current techniques for EV isolation and the need for standardized production protocols.

Therefore, it is essential to obtain pure and highly concentrated EVs for clinical use. EVs are typically purified from human cells to ensure their safe use in the human body. However, this method is difficult to scale-up. Therefore, milk and plant EVs have attracted considerable interest as possible options for EV production at high concentrations. Currently, there is a lack of information regarding specific markers, proteins, and RNAs present in non-mammalian EVs. Thus, caution should be exercised when using these EVs as they may cause unexpected effects. To expand the number of EVs, it is necessary to establish a human-derived cell line and continue to characterize non-mammalian-derived EVs.

Numerous techniques have been proposed for EV purification; however, the most effective technique remains unclear. Obtaining high-purity EVs using only one method can be challenging because EVs must be separated from other extracellular vesicles based on their differences in size and density. While TFF purification and density gradient ultracentrifugation are recommended, other methods, such as spin-down filtration and size-exclusion chromatography, can also be used. It is important to continue discussing and exploring different options.

After purifying the concentrated EVs with high purity, it is essential to confirm that each EV contains a molecule with efficacy. This means determining the ratio of effective EVs to purified EVs and establishing a method for producing effective EVs efficiently and uniformly. The heterogeneity of EVs presents a significant challenge in this process, and it is necessary to develop a characterization method to address this issue. Currently, most EV characterizations are conducted using bulk assays owing to their small size. To overcome this heterogeneity, the development of single-molecule characterization methods is urgently required.

In our exploration of clinical trials, we conducted a cursory examination of EV applications, specifically their deployment as biomarkers in diagnostic contexts and their burgeoning utility in therapeutic modalities, notably in modulating immune responses in pathologies such as SARS-CoV-2. This chapter further illuminates the emergent field of engineered EV research, delineating the complexities inherent in harnessing their therapeutic potential and underscoring the imperative for rigorous trial methodologies, along with establishing standards for EV characterization and production. However, it’s important to note that the majority of clinical trials to date have been predominantly limited to conventional EVs derived from natural sources.

Engineered EVs offer a promising avenue, as they are anticipated to exhibit enhanced efficacy in addressing complex clinical challenges. However, this potential is tempered by the need for more comprehensive studies addressing the heterogeneity of EVs, particularly in terms of their size, concentration, and purity. To this end, several studies are underway to refine EV production methods, enhance their purity, and develop standardized protocols and characterization techniques.

We expect that addressing these challenges at the fundamental research level will lead to significant progress in the clinical applications of EVs. Once their production and characterization are thoroughly optimized and standardized, the advancement of engineered EVs holds the potential for substantial achievements in the clinical realm, potentially revolutionizing the field of targeted therapeutics.

## 6. Conclusions and Future Directions

In conclusion, our comprehensive exploration of EVs underscores their potential in the biomedical field. EVs, derived from diverse sources such as human cell lines, milk, and plants, present novel opportunities for therapeutic and diagnostic applications. Their low immunogenicity and high biocompatibility, particularly when expressing functional proteins, make them ideal for targeted therapies. However, challenges in large-scale production, purification, and the need for consistent quality and efficacy remain. Current clinical trials focusing on naturally derived EVs have laid the foundation; expanding our understanding and capabilities in engineering EVs is essential for maximizing their therapeutic potential. However, the broad clinical application of EVs is limited by technical drawbacks that fail to address heterogeneity and a lack of understanding of EV biogenesis. Merely employing active loading methods for loading drugs and proteins into EVs makes it challenging to prove the loading mechanism and quantitatively evaluate the loaded amount. Therefore, it is essential to intrinsically load drugs and proteins into EVs based on biological knowledge and quantitatively evaluate them. Particularly for the clinical application of EVs, it is crucial to characterize them at the single-molecule level.

Looking forward, the field of EV research is at a critical juncture. The successful translation of EV-based therapies from the laboratory to clinical settings hinges on overcoming heterogeneity, scale-up, and precise targeting challenges. With continued research progress, engineered EVs will revolutionize treatment modalities across various diseases. Embracing a multidisciplinary approach, which includes advanced manufacturing techniques, rigorous clinical trials, and innovative characterization methods, will be key to unlocking the full potential of EVs. The journey of EVs from a biological curiosity to a cornerstone of therapeutic innovation exemplifies the dynamic and evolving nature of the medical sciences.

## Figures and Tables

**Figure 1 pharmaceutics-16-00311-f001:**
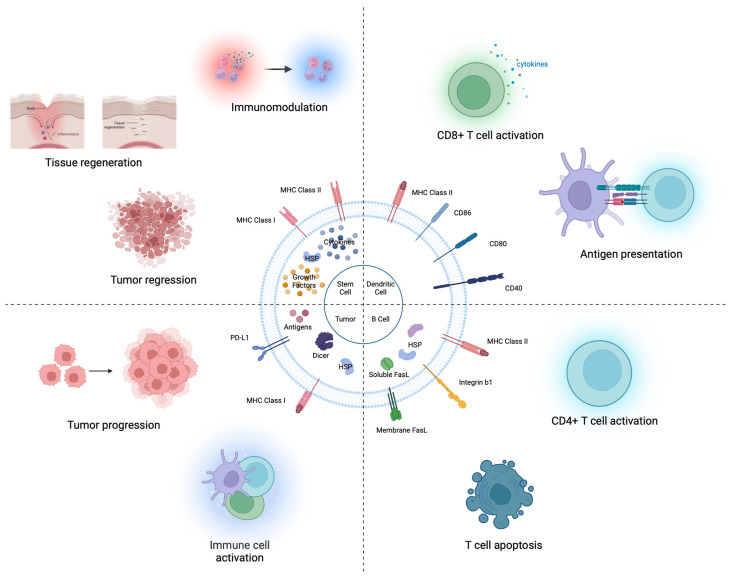
Intact EV therapeutics (X-derived EV). Stem cell-derived EVs contain various bioactive molecules (growth factors, cytokines, and heat shock proteins). These EVs exert effects on tumor regression, tissue regeneration, and immune modulation. DC-derived EVs contain antigen-presenting proteins (MHC I or II complexes). Mature DC-derived EVs can directly stimulate CD8 T cells and can activate CD4 T cells. B cell-derived EVs have MHC class proteins. B cell-derived EVs activate CD4 T cells and induce T cell apoptosis. Tumor-derived EVs harbor numerous molecules influencing immune activation (i.e., MHC class I, Antigens, and HSPs) or suppression (e.g., PD-L1, IL-10).

**Figure 2 pharmaceutics-16-00311-f002:**
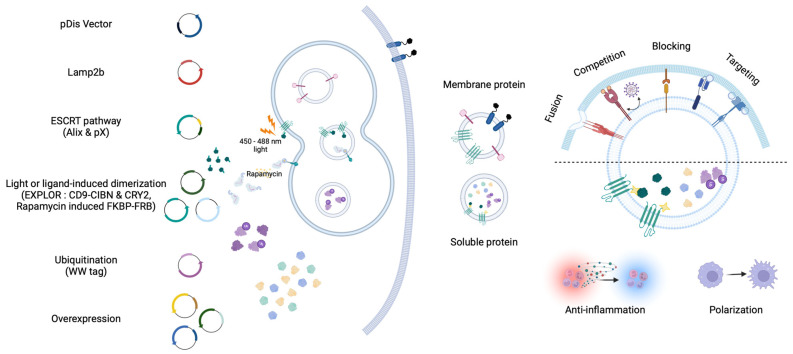
EVs that contain proteins with therapeutic effects on recipient cells. Membrane proteins are known to be well transported into EVs through EV biogenesis pathways (i.e., Lamp2b and Tetraspanin family). Proteins exposed on the surface of EVs are involved in the interaction with membrane proteins of recipient cells. The expression of membrane proteins in EVs encompasses fusion, competition, blocking, and targeting activities. Using the virus-derived fusion protein (VSVG), soluble proteins within EVs can be effectively delivered; however, certain membrane proteins can compete with virus proteins (i.e., IFITM and, ACE2) to reduce infection efficiency. Proteins involved in interactions with tumor and immune cells (i.e., CD47 and SIRPα) can be delivered via EVs, thereby blocking to enhance immune responses. Exposing specific peptides (i.e., RVG, GE11, and nanobody) can elevate cell-specific delivery capabilities by targeting recipient cells. The delivery of the target soluble proteins of interest into EVs can be achieved through concentration-dependent methods and active sorting. The active sorting approach utilizes EV biogenesis pathways (i.e., Lamp2b, ESCRT pathway, and ubiquitination, EXPLOR, Rapamycin induced FKBP-FRB) to effectively load EVs. These soluble proteins can be taken up by recipient cells, altering cellular activity or changing the microenvironment (i.e., anti-inflammation and polarization).

**Figure 3 pharmaceutics-16-00311-f003:**
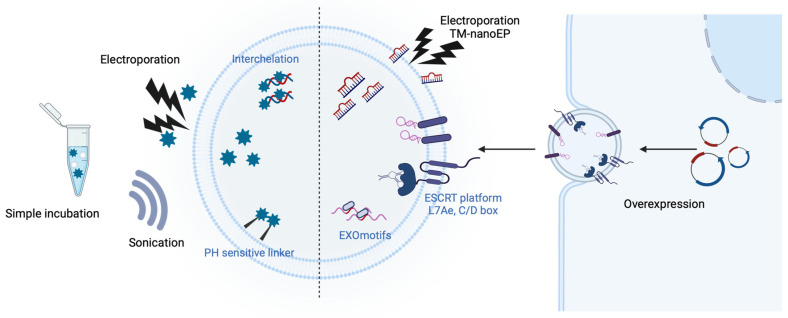
Drug and nucleic acids containing EVs. Techniques have been developed to load drugs and nucleic acids into EVs to reduce degradation, maximize the effectiveness of low doses, and minimize toxicity. For drugs, simple incubation with EVs or intercalation, as well as the use of pH-sensitive linkers, could be employed. Nucleic acids can be loaded into EVs via extrinsic methods (i.e., electroporation including track-etched membrane-nanoelectroporation (TM-nanoEP) and sonication) or intrinsic methods (i.e., motifs, ESCRT platforms, and overexpression). These methods can selectively deliver EV substances to recipient cells by exposing targeting proteins on the membrane proteins.

**Table 1 pharmaceutics-16-00311-t001:** Clinical trials related to EV-based therapy.

Functional Level	Cell Category	EV Origin	Key Signature	Target Disease	Phase	Status	NCT
X-derived EV	Stem Cells	Mesenchymal Stem Cells		Macular Holes	Early phase 1	Active, not recruiting	NCT03437759
				SARS-CoV-2	I/II	Completed	NCT04491240
				SARS-CoV-2	II	Enrolling by invitation	NCT04602442
				Multiple Organ Failure	Not applicable	Not yet recruiting	NCT04356300
				SARS-CoV-2	I/II	Not yet recruiting	NCT04798716
				Familial Hypercholesterolemia	I	Not yet recruiting	NCT05043181
				Osteoarthritis, Knee	I	Not yet recruiting	NCT05060107
				Severely infected children		Not yet recruiting	NCT04850469
				ARDS	I/II	Recruiting	NCT04602104
				Cerebrovascular Disorders	I/II	Recruiting	NCT03384433
				SARS-CoV-2	II/III	Recruiting	NCT05216562
				Alzheimer’s Disease	I/II	Recruiting	NCT04388982
				Refractory Depression, Anxiety Disorders, Neurodegenerative Diseases		Suspended	NCT04202770
				Diabetes Mellitus Type 1	II/III	Unknown status	NCT02138331
			Bone Marrow	SARS-CoV-2		Available	NCT04657458
			Bone Marrow, Exoplo	SARS-CoV-2	II	Completed	NCT04493242
			Bone Marrow, Exoplo	SARS-CoV-2	I/II	Not yet recruiting	NCT05116761
			Bone Marrow, Exoplo	SARS-CoV-2	II	Not yet recruiting	NCT05125562
			Bone Marrow, Exoplo	refractory Crohn’s disease	I	Not yet recruiting	NCT05130983
			Bone Marrow, AGLE 102	Burn Wounds	I	Not yet recruiting	NCT05078385
			Bone Marrow, AGLE 102	Epidermolysis Bullosa	I/II	Not yet recruiting	NCT04173650
			Synovial fluid-derived	Knee Injury	II	Recruiting	NCT05261360
			Allogenic Adipose	SARS-CoV-2	I	Completed	NCT04276987
			Allogenic Adipose	Healthy (safety and tolerance of aerosol inhalation)	I	Completed	NCT04313647
			Autogenous adipose tissue	Bone Loss	I	Not yet recruiting	NCT04998058
		Mesenchymal Precursor Cells	Human adipose derived	Pulmonary Infection	I/II	Recruiting	NCT04544215
		Umbilical Mesenchymal Stem Cells		chronic Graft Versus Host Diseases (cGVHD)	I/II	Recruiting	NCT04213248
		Stem Cells	UNEX-42	Bronchopulmonary Dysplasia	I	Terminated	NCT03857841
		Adipose Stem Cells		Periodontitis	Early phase I	Unknown status	NCT04270006
		Adipose-derived Stromal Cell		Osteoarthritis		Recruiting	NCT04223622
	Immune cells	Dendritic Cells	Peptide pulsed	Non-Small Cell Lung Cancer	II	Completed	NCT01159288
		T cells	Donor originated SARS-CoV-2 specific T cells	Corona Virus Infection Pneumonia	I/II	Active, not recruiting	NCT04389385
	Plant	Grape		Head and Neck Cancer Oral Musositis	I	Active, not recruiting	NCT01668849
		Citrus Limon		Metabolic Syndrome	Not applicable	Active, not recruiting	NCT04698447
		Ginger		Inflammatory Bowel Disease (IBD)		Recruiting	NCT04879810
		Aloe, Ginger		Polycystic Ovary Syndrome		Withdrawn	NCT03493984
Membrane protein delivery		Human embryonic kidney T-REx™-293	Overexpression CD24 protein	SARS-CoV-2	II	Active, not recruiting	NCT04969172
		Human embryonic kidney T-REx™-293	Overexpression CD24 protein	SARS-CoV-2	I	Recruiting	NCT04747574
Chemical drugs		Plant	Curcumin	Colon Cancer	I	Recruiting	NCT01294072
Nucleic acid delivery		Mesenchymal Stem Cells	KRAS G12D siRNA	Pancreatic Cancer with KrasG12D mutation	I	Recruiting	NCT03608631

## Data Availability

The data presented in this study are available in this article.

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
