# Peer review of "Extracellular Vesicles in Therapeutics: A Comprehensive Review on Applications, Challenges, and Clinical Progress"

_pharmaceutics, 2024, doi:10.3390/pharmaceutics16030311_

Round 1

Reviewer 1 Report

Comments and Suggestions for Authors

In this review, the authors review the use of cell-derived exosomes and highlight the potential of therapeutic exosomes and the requirements for successful clinical trials, including methods for large-scale production and purification. Overall, this a well-written review and covers many interesting pieces. However, there are several points of concern related to the exosome’s definition and applications. I would like to recommend a major revision before acceptance.

1.     Please review MISEV2018 for an accurate definition of small extracellular vesicles/exosomes.  (doi: 10.1080/20013078.2018.1535750)

2.     Session 2.3.2. Nucleic acids. Author primarily highlighted small RNAs. Please also include other relevant components like mRNA, which could play more important roles during diagnosis or treatment. Several recent papers have mentioned mRNA-Exosomes (10.1016/j.vesic.2022.100002; doi.org/10.1002/advs.202302622). They could be valuable to be involved in this review.

3.     For protein delivery, Some Cas9 RNP can be introduced.

4.     Some figure demonstration of recent exosome engineering technologies could be beneficial to the review.

Comments on the Quality of English Language

Minor editing of English language required

Reviewer 2 Report

Comments and Suggestions for Authors

Specific comments

In the introduction, authors should consider first introduce extracellular vesicles (EVs) in general, the different EV subtypes, including exosomes, microvesicles and apoptotic bodies, and their characteristics, before delving into exosomes. It may also be necessary to mention that the current EV preparations including exosome preparations are heterogenous with undetermined purity and undefined biogenesis origin. In view of these issues, the International Society for Extracellular Vesicles (ISEV) recommended the use of term “EV” instead of “exosome” in the MISEV2018 position paper.

In the conclusions and future directions, it is worthwhile to also discuss the challenges associated with engineering exosomes. These include issues of EV heterogeneity, loading efficiency with drug loading, luminal loading instead of mere association with the exosome membranes, and standards, etc.

This review is too long. Authors may want to refer to recent reviews for certain sections such as 3.2 purification including ultra-centrifugation, size-exclusion chromatography, tangential flow filtration, and asymmetric flow filed-flow fractionation.  

In 3.1.1 Human cell line - While there are concerns of tumorigenicity associated with TERT, there are other means of immortalizing the mesenchymal stem cells such as MYC transformation, and that have been reported to yield MYC-transformed MSCs with minimal tumorigenic potential and exosomes with comparable properties as those derived from the native MSCs.

Round 2

Reviewer 1 Report

Comments and Suggestions for Authors

The authors have satisfactorily addressed my concerns.